# Multilineage-Differentiating Stress-Enduring Cells (Muse Cells): An Easily Accessible, Pluripotent Stem Cell Niche with Unique and Powerful Properties for Multiple Regenerative Medicine Applications

**DOI:** 10.3390/biomedicines11061587

**Published:** 2023-05-30

**Authors:** Riccardo Ossanna, Sheila Veronese, Lindsey Alejandra Quintero Sierra, Anita Conti, Giamaica Conti, Andrea Sbarbati

**Affiliations:** Department of Neuroscience, Biomedicine, and Movement Sciences, University of Verona, 37124 Verona, Italy; sheila.veronese@univr.it (S.V.); lindseyalejandra.quinterosierra@univr.it (L.A.Q.S.); anita.conti@univr.it (A.C.); giamaica.conti@univr.it (G.C.); andrea.sbarbati@univr.it (A.S.)

**Keywords:** cell-based therapy, regenerative medicine, pluripotency, mesenchymal stem cells, stromal vascular fraction

## Abstract

Cell-based therapy in regenerative medicine is a powerful tool that can be used both to restore various cells lost in a wide range of human disorders and in renewal processes. Stem cells show promise for universal use in clinical medicine, potentially enabling the regeneration of numerous organs and tissues in the human body. This is possible due to their self-renewal, mature cell differentiation, and factors release. To date, pluripotent stem cells seem to be the most promising. Recently, a novel stem cell niche, called multilineage-differentiating stress-enduring (Muse) cells, is emerging. These cells are of particular interest because they are pluripotent and are found in adult human mesenchymal tissues. Thanks to this, they can produce cells representative of all three germ layers. Furthermore, they can be easily harvested from fat and isolated from the mesenchymal stem cells. This makes them very promising, allowing autologous treatments and avoiding the problems of rejection typical of transplants. Muse cells have recently been employed, with encouraging results, in numerous preclinical studies performed to test their efficacy in the treatment of various pathologies. This review aimed to (1) highlight the specific potential of Muse cells and provide a better understanding of this niche and (2) originate the first organized review of already tested applications of Muse cells in regenerative medicine. The obtained results could be useful to extend the possible therapeutic applications of disease healing.

## 1. Introduction

In the last decade, particular attention has been given to the study of different types of cells and to their application to favor and promote regenerative processes and for the cell-based treatment of many diseases [1,2]. In particular, great interest in research has been aroused by stem cells; their self-renewal capacity and their ability to differentiate into mature adult cells make them promising for human tissue regeneration [3,4]. Indeed, stem cells and their differentiated derivates are increasingly used in a number of cell-based therapies and are being studied worldwide, with the hope of healing the cell loss condition correlated to various diseases [5].

Innovative cell therapies can be used to restore cells lost in many human disorders, such as Parkinson’s disease, Crohn’s disease, pulmonary fibrosis, and diabetes mellitus, and to enhance the repair processes of various tissues [2,6]. Specifically, the application of stem cells in cell-based therapy promotes the repair response of diseased, dysfunctional, or injured tissue. This is obtained by using both the stem cell progenitor, which in vitro differentiates into tissue-specific cells, and their derivate (subpopulation of mature cells derived from stem cell differentiation in vitro) [2]. As shown in Figure 1, the differentiation capacity of stem cells depends on their specification potential.

Among all the stem cells, great importance is given to pluripotent stem cells, which include embryonic stem cells (ESCs), induced pluripotent stem cells (iPSCs) and Multilineage-differentiating stress-enduring cells (Muse cells). These groups of cells have the unique ability to differentiate into mature cells, originating from all three embryonic germ layers (Figure 1).

Totipotent stem cells generate all the cell types in an organism (e.g., zygote or fertilized egg), and pluripotent stem cells produce all the embryonic germ layers (endoderm, ectoderm, and mesoderm). The main pluripotent example is represented by embryonic stem cells (ESCs), which derive from the inner cell mass of the blastocyst [7,8]. Induced pluripotent stem cells (iPSCs) are reprogrammed from differentiated adult cells and exhibit pluripotent capacities.

One promising group of stem cells is a multipotent group of cells named mesenchymal stem cells (MSCs) or stromal stem cells. MSCs are adult stem cells isolated from mesenchymal tissues. They have the property of being able to differentiate into many other cell types [1,2]. In humans, MSCs sources include bone marrow, adipose tissue, umbilical cord tissue, dermis, and peripheral blood [9]. The fact that these MSCs are adipose tissue-resident cells makes them potentially attractive as they could be extracted and isolated from easily harvested fat by liposuction [1,6]. The application of MSCs in regenerative medicine presents two key advantages: (i) a low risk of tumorigenesis and (ii) a high differentiation capacity, which permits their application to the restoration of various pathological cell loss conditions [1,6,10]. Moreover, due to their origin from mesodermal lineage, they are able to differentiate into cells such as osteocytes, chondrocytes, adipocytes, skeletal muscle cells, endothelial cells, and cardiomyocytes. Therefore, they can find numerous fields of application in regenerative therapy studies [11].

Generally, adult stem cell renewal is limited because they can only differentiate into specific cells of a single tissue. Finding stem cells able to differentiate into all types of tissue is a challenge. ESCs have emerged as the gold standard of pluripotent stem cells and the class of stem cells with the highest potential for contribution to regenerative and therapeutic applications [12], but the process through which ESCs are obtained contrasts with principles of an ethical nature [7]. A possible way to overcome this obstacle was already found in 2006 when Takahashi and Yamanaka published the results of their work on reprogramming mouse embryonic fibroblasts into a new type of cell, iPSCs, which exhibited morphology, growth properties, and marker gene expression of ESCs, without ethical concerns. iPSCs can differentiate in various human tissues and exploit regenerate properties [1,2], given their pluripotency capability. They are widely used to obtain stem cells with regenerative potential [12]. Unlike human ESCs, iPSCs raise no ethical concern regarding the onset of human personhood [3,5]. Unfortunately, with the use of these cells, there is the important risk of tumor formation, risk correlates with both pluripotency and self-renewal, and this has emerged as a critical factor in determining the pluripotent capacities of both ESCs and iPSCs [11].

The production of iPSCs represents a breakthrough in regenerative medicine, as it provides new opportunities for understanding basic molecular mechanisms of human development and molecular aspects of degenerative diseases [3,13]. However, iPSCs still present some technical issues related to immune rejection after transplantation and the mentioned potential tumorigenicity, which means that additional steps need to be considered before using iPSCs as a viable tool for in vivo tissue regeneration [5,14].

In recent years, a new group of cells with broad differentiation capacity has been discovered, in addition to the MSCs mesodermal lineage. This mesenchymal stem cells’ subpopulation is a species of stress-tolerant stem cells: the multilineage-differentiating stress enduring cells, or Muse cells [15], first described in 2010 [6]. Unlike normal MSCs, which differentiate into the mesodermal lineage, this peculiar cell subpopulation has been proven to be able to differentiate into the other two embryonic germ cell lineages, the endodermal and ectodermal [1,10,16]. Consequently, Muse cells are particularly promising, as they could potentially regenerate every type of tissue of the human body.

Considering all the functional characteristics of Muse cells, we performed a systemic review to better understand and thoroughly investigate their action mechanism, and their regenerative effect, based both on their broad differentiation capacity and on their single properties. Moreover, all the Muse cells treatments already tested in regenerative applications and used for the resolution/improvement of various pathological conditions were taken into consideration.

## 2. Bibliographic Research

A database search was conducted in Medline (PubMed) across all articles published up to December 2022. The subject heading used for the initial search was “multilineage-differentiating stress-enduring cells”. The papers included in this review were selected using the PRISMA 2020 inclusion method. Papers for which the full text was not available, letters, meetings, and conference abstracts were excluded. Considering the remaining studies, papers with no thematic relevance for the discovery, characterization, and application of Muse cells were also excluded. Ninety-five articles were identified with the search term “multilineage-differentiating stress-enduring cells”. Forty-nine articles met the inclusion criteria and were included. The second bibliographic research was: “stem cells in regenerative medicine”, “fat harvesting”, “stromal vascular fraction”, “brain lacunar stroke”, “bladder inflammation”, “cerebral ischemia”, “intracerebral hemorrhage”, “malignant gliomas”, “spinal cord injury”, “lung ischemia”, “diabetes”, “diabetic skin ulcers”, “myocardial infarction”, “aortic aneurysm”, “skin regeneration”, “corneal scarring”, “osteochondral lesions”, “intestinal epithelium injuries”, “liver dysfunction”, and only the full text and topic inherence papers were included.

The PRISMA 2020 inclusion/exclusion criteria were used to standardize the articles included, and the PRISMA 2020 checklist was used to examine the text of the articles and correct or add any missed key points.

As shown in Figure 2, the selected articles were divided as follows: 49 studies were focused on Muse cells, 9 on stem cells and regenerative medicine, 6 on fat harvesting, 4 on the stromal vascular fraction, and 18 on specific pathologies for which Muse cells were used as therapy.

## 3. Muse Cells

### 3.1. Basic Characteristics and Isolation Methods

Muse cells are peculiar stem cells found in adult human mesenchymal tissues, i.e., bone marrow, adipose tissue, dermis, and peripheral blood [9] (Figure 3). Therefore, they are easily accessible, and their sampling is minimally invasive compared to other stem cells. The main substantial feature of Muse cells is that they can produce representative cells of all three germ layers [4], starting from a single cell [10,15] and that they can also self-regenerate [4,17].

Muse cells are distinct stem cells found in adult human mesenchymal tissue, such as bone marrow, adipose tissue, dermis, and peripheral blood. They differentiate into cells of the three germ layers, such as hepatocytes, neurons, epidermal cells, adipocytes, osteocytes, chondrocytes, bile cells, insulin-producing cells, and functional melanin pigment-producing cells (melanocytes), while the other MSCs (non-Muse cells) differentiate only in cells of their mesenchymal lineage (adipocytes, osteocytes, and chondrocytes).

As well as various types of adult somatic cells, Muse cells can be differentiated into iPSCs when transduced with four specific transcription factors: Oct3/4, Sox2, Klf4, and c-Myc.

Unlike ESCs or iPSCs, Muse cells are not advantageously immortal in culture [18]. The in vitro non-immortality property translates into in vivo non-tumorigenicity [2,10,16]. It has been demonstrated that they do not form teratomas when transplanted into the testes of immunodeficient mice, contrary to what is often observed after the transplantation of ESCs or iPSCs [2]. The non-tumorigenic characteristics could make these cells a robust and safe tool, with low risk in preclinical and clinical uses and with essential advantages if compared with the other pluripotent stem cells [10,15].

Several authors have described and characterized Muse cells, focusing on stress tolerance and multilineage differentiation capacity [19]. Muse cells can be easily isolated from cultured ADSCs by applying stress conditions in the culture, or CD105/SSEA3 double positive cell sorting, using FACS technology. The first method to isolate Muse cells consists of MSCs extraction from mesenchymal tissues (such as bone marrow, dermis, umbilical cord, blood, or adipose tissue) (Figure 3). Subsequently, they have to be cultured for a long time under stressful conditions (long-term exposure to the proteolytic enzyme collagenase, serum deprivation, low temperature, and hypoxia) [10]. In this way, most of the MSCs die, and only the Muse cells survive [15]. The stress conditions resistance of Muse cells could be due to their characteristic non-homologous end joining (NHEJ) system to survive solid genotoxic stress [20]. The second method to isolate Muse cells from cultured MSCs is the fluorescence-activated cell sorter, in which the specific SSEA3 and CD105 double-positive cells are isolated since Muse cells express both these markers [21]. The SSEA3 marker identifies pluripotent stem cells. In addition, Muse cells express mesenchymal markers, such as CD29, CD90, and CD105. So, they can be identified as double-positive cells for mesenchymal and pluripotency markers [15].

Sorted Muse cells can grow under both suspension and adhesion culture conditions, with some phenotypic behavior differences. Suspension-cultured Muse cells tend to form embryonic-like cell clusters and maintain their capacity for self-renewal and their pluripotent gene expression (triploblastic differentiation). Adhesion-cultured Muse cells tend to lose these staminality characteristics [20].

### 3.2. Muse Cells Pluripotency and Their Unique Regenerative Features

As known, MSCs can usually differentiate into mesenchymal tissues thanks to their multipotency [10,15]. Given their MSCs source, also Muse cells can differentiate in mesenchymal tissues. However, Muse cells have been reported to present a higher differentiation rate (80–95%) than other MSCs. This may depend on the fact that Muse cells express pluripotent genes such as OCT3/4 [19]. However, MSCs of different origins have shown the potential for neural differentiation [22,23]. In view of this, Muse cells are very versatile and can give rise to not only mesodermal cells (osteocytes, chondrocytes, and adipocytes) but also to ectodermal and endodermal cells [2,14,16], such as hepatocytes [24], neurons [24,25], bile cells, insulin-producing cells, and functional melanin pigment-producing cells (melanocytes) [24] (Figure 3).

In addition to pluripotent properties, Muse cells present very interesting characteristics. First of all, in vivo, Muse cells were delineated by high tropism for damaged tissues. This aspect, together with their pluripotency, gives them the property to differentiate into tissue-specific marker-expressing cells. This depends on the stimulation received by the damaged tissue into which they have migrated [2] after intravenous injection or on the endogenous recruitment of host cells. In this way, they contribute to damaged tissue regeneration. Consequently, Muse cells could be injected into any type of disease model in order to migrate to lesions, differentiate into specific cells subtype, and restore cells lost after pathological conditions. Therefore, in the context of tissue regeneration, they have the potential to make critical contributions [10].

During pathophysiological events, the immune-inflammatory response could counteract and inhibit healing processes. In this context, previous studies have highlighted fascinating intrinsic immunomodulatory properties of Muse cells with anti-apoptotic and anti-fibrotic effects [19]. In the same manner as other cells of regenerative niches [26], Muse cells reside in mesenchymal tissues and are close to blood vessels that carry immune cells and, consequently, modulate immune-inflammatory response. Their immunomodulatory capacity was first reported by Gimeno et al. and Alessio et al. and was described as a decrease of proinflammatory TNF-α in a mouse macrophage-like cell line in vitro [27]. This reduction caused the loss of antigen-specific stimulation of Th1-type cytokines (IFN-γ and TNF-α) and led to the reduction of inflammation.

Moreover, TGF-β, a key immunosuppressive cytokine, is a well-known inflammatory downregulator and a tissue regenerator promoter [28]. As it is highly expressed by Muse cells [27], this suggests intrinsic anti-inflammatory and pro-reparatory properties displayed from these cells. Finally, Muse cells may have a key role in cell survival by influencing anti-apoptotic activities and regulating DNA damage checkpoint and ERK5 signaling pathways [29].

## 4. Applications of Muse Cells in Regenerative Medicine and Disorders Healing

Given the described potentiality, Muse cells have aroused wide scientific interest. In fact, they have been employed in preclinical studies of regenerative medicine to treat various diseases and for tissue regeneration. They have been used to treat damage to various organs, exploiting their differentiating ability to restore the cell lost after pathological conditions. Recently, Muse cells have been studied as a cell-based therapy to renew tissues in different medical applications (Figure 4).

The medical applications of Muse cells include studies of their regeneration properties of skin, liver, muscle, brain, lung, intestine, β-pancreatic cells, cornea, myocardium, bone, and cartilage. Muse cells are used in many preclinical studies of pathologies (e.g., cerebral or lung ischemia, diabetes, aortic aneurysms, etc.).

In particular, Muse cells were tested in various clinical and preclinical studies aimed at treating different pathological conditions:Acute lung ischemia;Aortic aneurysm;Bladder inflammation;Brain lacunar stroke;Cerebral ischemia;Corneal scarring;Diabetes;Intestinal epithelium injuries;Intracerebral hemorrhage;Liver dysfunction;Malignant gliomas;Myocardial infarction;Osteochondral lesions;Skin damage;Skin ulcers;Spinal cord injuries.

### 4.1. Acute Lung Ischemia

Acute Lung Ischemia is a disorder that manifests clinically through pulmonary edema, respiratory distress, and hypoxemia [30]. Rat models of ischemic lung injury were infused with 1.5 × 10^5^ Muse cells and with non-Muse cells 2 h after left pulmonary ischemia induction by left pulmonary artery occlusion. Infused Muse cells remained in the injured lung more efficiently than non-Muse cells due to their lesion tropism. Injured lungs infused with Muse cells stimulated higher levels of biomarkers related to tissue repair, apoptosis prevention, and alveolar fluid clearance [31]. This suggests that Muse cells may protect lung functions and structure from ischemic injury.

### 4.2. Aortic Aneurysm

Aortic aneurysm (AA) is a permanent and irreversible localized dilatation of the aorta and results from changes in the aortic wall structure, including thinning of the media and adventitia. This is due to the loss of vascular smooth muscle and endothelial cells and the degradation of the extracellular matrix [32]. Muse cells treatment of AAs was proposed as a smooth muscle and endothelial cells regeneration therapy by Hosoyama et al. Human bone marrow Muse cells were injected intravenously at day 0, day 7, and 2 weeks (2 × 10^4^ cells/mouse) after inducing AAs by the periaortic incubation of CaCl_2_ and elastase in severe combined immunodeficient mice. At 8 weeks after aortic damage induction, infusion of human Muse cells attenuated both aneurysm dilation and size, compared to the control and non-Muse treated groups. Histological analysis confirmed the spontaneous differentiation of Muse cells into endothelial cells and vascular smooth muscle cells, favoring aneurysm healing [33].

### 4.3. Bladder Inflammation

Bladder inflammation is a condition with chronic pelvic pain, pressure, or discomfort that is perceived to be related to the bladder. It is accompanied by other urinary symptoms, such as a persistent urge to void or increased urinary frequency [34]. A rat bladder inflammation model was created by Furuta et al., and Muse cells were injected into the anterior and posterior bladder walls using a microsyringe. This resulted in a significant amelioration in urinary frequency and bladder capacity [35].

### 4.4. Brain Lacunar Stroke

Brain Lacunar Stroke (BLS) is caused by a non-thrombotic obstruction of small arteries supplying the deep cortical structures, causing ischemic damage. As the second leading cause of death worldwide, stroke has a large social impact [36]. Two animal model studies have reported the use of Muse cells in BLS. In a first study by Abe et al., human-derived Muse cells were administered in immunodeficient murine lacunar stroke models by injection into the cervical vein, at different doses, during both the subacute and chronic phases of stroke. At 22 weeks, tumorigenesis and adverse effects were not detected. Interestingly, the high-doses treated mice presented a more significant functional recovery compared to the low-dose treated mice, both in the subacute and chronic phases of the disease. Histological analysis revealed that human Muse cells were mainly distributed in the peri-infarct area and expressed neuronal and neurogenic markers (neuronal nuclei, NeuN, and microtubule-associated protein-2, MAP-2) [37]. In a second study by Uchida et al., Muse cells were directly injected into the peri-injured brain in an immunodeficient mice model of subacute Lacunar Stroke. At 8 weeks post-injection, the transplanted Muse cells remained in the host brain and were spontaneously differentiated into cells expressing NeuN and MAP2, indicating proper neuronal differentiation and regeneration. Muse-transplanted stroke mice displayed significant recovery, with no evidence of tumor formation [38]. These studies evidence their potential for neural regeneration [37].

### 4.5. Cerebral Ischemia

Cerebral ischemia, a common form of stroke, is the fifth leading cause of death and disability worldwide. It is caused by the blockage of a blood vessel due to a thrombus or embolism, resulting in ischemic damage [39]. The study of Muse cells application in this disease was performed in a mouse model. Immunodeficient mice with permanent middle cerebral artery occlusion received human-Muse cells transplantation into the ipsilateral striatum at 7 days post occlusion. Functional mice recovery became evident 35 days after transplantation, compared to controls. Immunohistochemistry analysis revealed Muse cells integrated into the peri-infarct cortex and differentiated into NeuN-expressing cells. These findings confirm the ability of integrated Muse cells to spontaneously differentiate into neuronal marker-positive cells, consequently favoring functional recovery after ischemic stroke [40]. Muse cells were also tested in phase 1 human trial in neonatal ischaemic encephalopathy. This study is still ongoing [41].

### 4.6. Corneal Scarring

Corneal scarring (CS) usually occurs after injury, infection, or surgery. Corneal transplants require donor tissue and long-term medical care and can lead to transplant rejection. Therefore, more effective approaches are needed to prevent or improve corneal fibrosis [42]. Stem cell therapy holds promise for the treatment of CS. In this context, Muse cells were isolated from the lipoaspirate and expanded approximately 100-fold into Muse spheroids as a first approach. These activated Muse spheroids enabled differentiation into corneal stromal cells, expressing their characteristic marker genes and proteins.

In a second strategy, direct implantation of differentiated Muse cells into corneal stromal cells in injured corneas was performed in mice. The implantation prevented the formation of CS, increased corneal re-epithelialization and nerve regrowth, and reduced the severity of corneal inflammation and neovascularization. Thus, Muse cell therapy seems to be a promising avenue for developing therapies that can heal CS [43].

### 4.7. Diabetes

Diabetes is a metabolic disease characterized by high blood sugar levels for an extended period [44]. Type 1 diabetes is an autoimmune disease characterized by the destruction of pancreatic insulin-producing β cells mediated by autoreactive T lymphocytes and pro-inflammatory agents. Muse cells secrete significant amounts of TGF-β1, a key cytokine that regulates the down-modulation of T lymphocytes and macrophages. Moreover, they show the in vitro ability to differentiate into insulin-producing cells, replacing the β cells lost in this pathology [45]. Ali Fouad et al. tested the release of insulin by Muse and non-Muse cells upon exposure to increased glucose concentration. The amount of insulin released from Muse cells gradually increased in response to increasing glucose concentrations and was higher compared to non-Muse cells [46]. Considering the β pancreatic regenerative potential of Muse cells, their administration in diabetic NOD mice (non-obese diabetic mice) was tested as a model of spontaneous autoimmune diabetes. Diabetic animals received a single intraperitoneal injection of 1 × 10^6^ Muse cells (extracted and isolated from adipose tissue). The glucose levels of the control group increased dramatically, reaching high blood glucose levels after one week (>500 mg/dL), while the group treated with Muse cells showed blood glucose levels ranging from 202 to 500 mg/dL during a 7-week observation period. Muse cells were able to control the autoimmune process after the onset of spontaneous diabetes [45]. Further studies are necessary to understand the migration process of Muse cells to damaged pancreatic β cells. It would also be interesting to investigate whether they are able to differentiate in vivo into insulin-producing cells and whether TGF-β Muse cell production could modulate the autoimmune attack of T lymphocytes and macrophages against pancreatic cells.

### 4.8. Intestinal Epithelium Injuries

Lesions of the intestinal epithelium can impair the three critical functions of the intestinal lumen: nutrient digestion and absorption, protection, and hormone secretion [47]. The ability of Muse cells to resolve these lesions was tested in vitro by Sun et al. Lesions of intestinal epithelial crypt cell 6 (IEC-6) and colorectal adenocarcinoma 2 (Caco-2) cells were induced by stimulation of tumor necrosis factor-α. Muse cells (5 × 10^5^ /well) were co-cultured with IEC-6 and Caco-2, showing significant protective effects on the intestinal barrier structure. The underlying mechanisms were related to reduced levels of interleukin-6 (IL-6), and interferon-γ (IFN-γ), and restoration of transforming growth factor-β (TGF-β), and interleukin-10 (IL-10) in the inflammation microenvironment [48].

### 4.9. Intracerebral Haemorrhage

Intracerebral hemorrhage (IH) is the deadliest subtype of stroke. There is currently no effective treatment. It is usually provoked by head trauma [49]. In the study by Shimamura et al., a mouse model of IH was obtained by injecting cardiac blood into the left putamen of immunodeficient mice. Five days later, 2 × 10^5^ human bone marrow-derived Muse cells were injected into the IH cavity. Three weeks after the transplant, the Muse-treated group showed a functional motor recovery compared to controls. Moreover, the survival rate of the treated mice was significantly higher than that of the control mice. After 10 weeks, the injected Muse cells showed positivity for neural NeuN and neurogenic MAP-2 markers, suggesting that direct injection of Muse cells into the hematoma cavity resulted in the long-term survival of the transplanted cells, spontaneous differentiation into neuronal marker-positive cells, and the rapid improvement of the neurological deficit [50].

### 4.10. Liver Dysfunction

Liver dysfunction is attributed to a lack of sufficient functional cells. Liver cirrhosis is the last common pathological pathway of liver injury and arises from a wide variety of chronic liver diseases [51,52]. Katagiri et al. intravenously infused GFP-labelled human Muse cells into a mouse model of physical partial hepatectomy. Muse cell integration was observed in the damaged area adjacent to the transection line, wherein GFP-labelled Muse cells appeared in the periportal regions adjacent to the actual lesion 1 week after hepatectomy. After 2 weeks of infusion, some of the Muse cells started forming bile duct-like structures. Muse cells were also found in the sinusoid area. Supplementation of Muse cells monitored for up to 4 weeks showed that the population of Muse cells that had integrated into the liver included choanocytes (17.7%), hepatocytes (74.3%), Kupffer cells (6.0%), and sinusoidal endothelial cells (2.0%) [52].

In another study, a 70% piglet hepatectomy model was established, and 1 × 10^7^ of allogeneic-Muse cells were injected. As a result, Muse cell administration contributed to the decrease in hyperbilirubinemia. Allogeneic Muse cells delivered via the portal vein were integrated into the remnant liver and expressed hepatocyte markers. In addition, Muse cell administration suppressed the inflammatory reaction and necrosis in the liver necrosis areas [53].

### 4.11. Malignant Gliomas

Malignant gliomas (MG) are primary tumors that arise in the central nervous system and for which a limited range of therapies exists [54]. MG have high morbidity and mortality. Hence, effective treatment could be of great importance [20]. Yamasaki et al. intracranially injected Muse-tk (Muse cells transfected with the suicide gene Herpes Simplex Virus thymidine kinase, HSVtk) into immunodeficient nude mice with human MG. After cell injection, Muse-tk cells showed potent migratory activity toward glioma cells. The glioma significantly reduced in size within 2 weeks of injection. The number of human glioma cells was markedly reduced in the nude mouse brains, leading to the inhibition of tumoral mass growth [55]. These findings support the possible Muse cells application in tumor targeting and elimination.

### 4.12. Myocardial Infarction

Myocardial infarction (MI) is mainly caused by a decrease or interruption of blood flow to a portion of the heart, leading to necrosis of the heart muscle [56]. The myocardium itself does not have a high regenerative capacity, and it is important to minimize the loss of cardiomyocytes and maintain cardiac function after MI. The scientific community is studying myocardial tissue regeneration by transplanting bone marrow stem cells. Recently, it was reported that Muse cells could reduce the MI size and improve cardiac function via cardiomyocytes and vessel regeneration and have paracrine effects compared with canonical MSCs [57]. In particular, in the first human study of Muse treatment, intravenous injection of 1.5 × 10^7^ Muse cells markedly reduced the MI size, improved left ventricular function, and attenuated cardiac remodeling through regeneration of functional cardiomyocytes and vessels, and with paracrine effects [58]. Muse cells were also tested in a Mini-pigs MI model, in which animals received coronary artery occlusion followed by 2 weeks of reperfusion. Human Muse cells (1 × 10^7^) were administered intravenously 24 h after reperfusion. The results were intriguing: MI size was significantly reduced in Muse-treated animals, and Muse cells were observed in the infarct area expressing cardiac and vascular markers [59].

### 4.13. Osteochondral Lesions

Osteochondral lesions are mainly associated with sprains and fractures related to several traumatic events. Many medical approaches to solving them are still being studied [60]. Mahmoud et al. created an osteochondral defect in the patellar groove of immunodeficient rats. Subsequently, 5 × 10^4^ Muse cells were injected into the lesion site. After 12 weeks, the Muse cell-treated group showed white repaired tissue with a predominantly smooth and homogenous surface, while no repair tissue was detected in the control group [61].

### 4.14. Skin Damage

Skin damage includes severe wounds or burns, which can progress to a chronic wound. Many chronic wounds fail to heal, resulting in amputations and death. The surgical procedures available for skin healing often have a limited supply of healthy donor tissue. The use of foreign tissue provides a substitute; however, it also poses a risk of infection and immune rejection. To overcome the limitations of surgical procedures in wound healing treatments, cell-based therapy may show promise [62]. Recent advances in stem cell biology, genome editing capabilities, and skin grafting techniques give rise to potential treatments of soft tissue defects [63]. Adipose-derived Muse cells, when cultured in a melanocyte-inducing medium, differentiate into melanocytes, which can be used for skin regeneration processes [21]. Hu et al. reconstituted skin using Muse cell-derived skin components (keratinocytes, melanocytes, and fibroblasts), demonstrating the effective generation of dermis components [63]. In another study, a mouse model of atopic dermatitis was treated with subcutaneous injection of Muse cells. Muse cells significantly alleviated scratching behavior in mice, reduced dermatitis, and played an active role in healing damaged skin [64].

### 4.15. Skin Ulcers

Skin ulcers (SUs), caused by vascular abnormalities, are a major cause of morbidity and mortality in diabetic patients [65]. In Sanches et al., SUs were generated in severe combined immunodeficiency mice with type 1 diabetes. Treatment with Muse cells significantly accelerated wound healing compared with treatment with non-Muse cells. Therefore, Muse cell treatment promoted stronger SU healing [16].

### 4.16. Spinal Cord Injury

Spinal cord injury (SCI) is a traumatic event that leads to impairment of motor and sensory functions. After the primary mechanical insult, the first consequence is neurodegeneration and the subsequent and consequent neural loss [66]. In the research of Kajitani et al., a rat model of thoracic SCI was intravenously administered (into the tail vein) with 3 × 10^5^ Muse cells. Two to eight weeks after injury, Muse-administered rats showed an improvement in the hind limbs’ locomotor function. Additionally, Muse cell injection prevented SCI. In fact, after SCI, spinal cord structural preservation was observed. Subsequently, Muse cells were differentiated into neural cells, resulting in the regeneration of the lesioned spinal cord parenchyma [67]. This supports Muse cells’ neural cell loss restoring properties in a spinal cord injury model.

## 5. Fat as an Optimal Source of Muse Cells Isolation

### 5.1. Fat Harvesting: An Established Procedure with Many Applications

Fat grafting is a consolidated procedure in reconstructive, aesthetic, and regenerative medicine due to the presence in the adipose tissue of a high concentration of MSCs [68,69]. The regenerative potential of adipose tissue is well known, and it is continuously updated thanks to numerous studies concerning the application of autologous adipose tissue transplantation in reconstructive and regenerative medicine [70,71,72]. The lower abdomen, buttocks, lateral femoral region, and inner side of the upper thighs are all ideal sites for fat harvesting due to the distribution of human adipose tissues and relatively large depots [73]. Harvesting human adipose tissue by lipo-aspiration is a safe and non-invasive procedure, and hundreds of millions of cells can be isolated from 1–2 L of lipoaspirate material [6,10]. Subsequent fat grafting is an established surgical technique used in plastic surgery to restore deficient tissue and, more recently, for the putative regenerative properties of fat [74].

The clinical applications of fat grafting are extensive. Among all, the most diffused are breast applications, buttock augmentation, facial contouring, hand rejuvenation, and fibrosis and scar reduction [75]. Healthy fat introduced into irradiated tissues of oncological subjects appeared to reverse radiation damage (fibrosis, scarring, contracture, and pain) [74].

There are also many studies supporting the efficacy of fat grafting in surgical applications such as in temporomandibular joint surgery, for the treatment and prevention of ankylosis, fibrosis, or heterotopic ossification in a total joint prosthesis; in neurosurgery, to treat, or prevent cerebrospinal fluid leaks in the spine; in otolaryngology for the ear obliteration, vocal cord surgery, and cleft lip and palate reconstruction [76].

The benefits of fat grafting applications in pathological conditions could be in part explained by the presence of MSCs subpopulations, such as the Muse cells. Restoration of tissue contour and volume has been attributed to adipogenic differentiation of fat-derived MSCs. The presence of new fat-derived cells at the fat graft site has been imputed to both the direct differentiation of introduced MSCs into adipocytes and the paracrine effects of MSCs to influence local stem cells to differentiate [74]. Moreover, adipose tissue could prove to be the ideal source for Muse cell isolation [4,6,10].

### 5.2. Fat Composition

The composition of harvested fat, such as fat derived from lipoaspirate, is schematized in Figure 5A. As shown in this figure, fat is composed of a heterogeneous cell subpopulation, such as adipocytes, adipose-derived stem cells (Muse and non-Muse cells), monocytes and macrophages, extracellular matrix components, pericytes, endothelial cells, and erythrocytes [77]. The Stromal Vascular Fraction (SVF) is constituted by the cellular components of fat. Then, the ADSCs reside in the SVF [78]. Altogether, the SVF contains a milieu of cells that constitute the functional cellular niche for regenerative processes [79,80]. MSCs are easily obtained from fat (also called ADSCs) and have similar properties to bone marrow-derived MSCs. Interestingly, adipose tissue contains a higher number of MSCs, compared to bone marrow [81]. So, it is a more effective source of stem cells (500 times greater when counted per unit volume than bone marrow) and without ethical concerns regarding its procurement [82]. The ADSCs content within the SVF of lipoaspirate samples is well known, and consequently, the SVF is used in several therapies and in a wide of human regenerative medicine approaches. The ADSCs it contains promote revascularisation, activate local stem cell niches, modulate immune responses via paracrine secretion of numerous bioactive molecules, promote wound healing, exhibit anti-inflammatory activity, and promote angiogenesis. These regenerative effects could be partially due to the MSCs and Muse cells’ content of SVF [83,84]. In fact, flow cytometry analysis confirmed that cells isolated from Bichat and abdominal adipose tissue specimens were classified as MSCs, due to the expression of CD105 and CD73. Moreover, a subpopulation was found simultaneously expressing SEEA3 and CD105, confirming the Muse cells’ presence in the ADSCs subpopulation [85].

### 5.3. Muse Cells Isolation from Adipose Tissue

Using fresh fat (from liposuction), Heneidi et al. and Simerman et al. developed a new methodology for the isolation of human Muse cells based on the application of severe cellular stress conditions to ADSCs, which induce the generation of human adipose-derived Muse cells (Muse-ATs) [4,10]. Muse-ATs harvested by lipoaspiration can also be isolated using the CD105/SSEA3 double-positive sorting of Muse cells.

Under physiological conditions, Muse-AT cells reside within the SVF of the adipose tissues. Hence, Muse isolation requires fat harvesting, SVF separation, and finally, the Muse cells’ isolation [11]. From a practical point of view, adipose tissue is minced and digested with or without collagenase (enzymatic or non-enzymatical digestion). The resulting cell suspension is filtered and centrifugated. The pellet-containing cells are cultured in Dulbecco’s modified Eagle’s medium (DMEM) supplemented with 10% fetal bovine serum and 1% penicillin/streptomycin. After 24 h, non-adherent cells are removed, while adherent cells are sub-cultured when they reach 70–80% confluency. In this way, an ADSCs culture is obtained. At this point, Muse cells can be isolated from non-Muse cells by cell sorting (using double positive CD105+/SSEA3+ cells by fluorescence-activated cell sorting, FACS) or by stress culture (Figure 5B), i.e., cultured with nutrient or oxygen deprivation, oxidative stress, and/or adverse temperature conditions [25].

## 6. Discussion and Conclusions

Thanks to their unique properties, stem cells promise to be universally used in clinical medicine, especially in the regeneration of many organs and tissues of the human body. This fascinating topic is receiving increasing attention from research world specialists from different branches of science and will undoubtedly present one of the most studied fields in the new millennium.

Multilineage-differentiating stress enduring cells, known as Muse cells, are stem cells living in mesenchymal tissues (such as adipose tissue) that exhibit pluripotent properties, such as triploblastic differentiation. Muse cells can differentiate into a diverse cell subpopulation, potentially with the ability to regenerate a diverse spectrum of tissues and organs [2,10,15,24,25]. Also, considering their additional features, such as immune response inhibition, in vitro non-tumorigenicity, and the ease with which they are harvested from liposuction, Muse cells appear particularly attractive and promising.

Current scientific knowledge about Muse cells reveals their beneficial effect and their possible application with positive results in a broad spectrum of diseases.

The promising Muse cell applications are also strengthened by their high tropism capacity for damaged tissues, toward which they spontaneously migrate and differentiate into tissue-specific cell subpopulations, contributing to the regeneration of damaged tissues. Hypothetically, due to their unique characteristics, Muse cells could be injected into any disease model in order to migrate to lesioned organs, differentiate into specific cells subtype, and restore cell loss after pathological conditions [16,21,31,40,48,55,58,61,63,67]. Moreover, Muse cells seem to have immunosuppressive properties [27], the ability to reduce inflammation and increase cell survival, and not to cause teratomas if injected, thanks to their non-immortal property [2,10,16].

Evaluating what are the limitations in the medical applications of Muse cells, firstly, cell-based therapies in medicine require their isolation and expansion from lipoaspirates (or other mesenchymal tissues); however, cell manipulation is considered illegal in many countries. Moreover, further characterizations of their homing in the organism, how they are recruited by injured tissues, and their action mechanisms are needed. Other limitations in their application could derive from the impossibility of harvesting adipose tissue from a patient, given by some pathological conditions or other causes. Furthermore, the basics of the mechanism of action of the regenerative effect of Muse cells should be thoroughly investigated. Indeed, other molecular factors and mechanisms and exosome secretion could contribute to the observed regenerative effects.

The high scientific interest and the great use of Muse cells in the medical and regenerative field, and the enormous potential that these stem cells seem to have, make further studies interesting to characterize their presence and abundance in different fat pads (abdomen, buttocks, leg, Bichat…) to improve the knowledge of their possible source and facilitate their autologous sampling.

Given that lipo-aspiration is a safe and non-invasive procedure, and given that Muse-AT is obtained and intended for autologous applications, where cell isolation requires a simple but highly efficient purification technique, and given that Muse-AT cells could provide an ideal source of pluripotent-like stem cells, Muse cells seem to be a very powerful tool for regenerating any type of tissue. Therefore, the results of ongoing clinical trials are awaited. Indeed, if these were comparable to those obtained with the application of other cell populations, the use of Muse cells could be reduced.

## Figures and Tables

**Figure 1 biomedicines-11-01587-f001:**
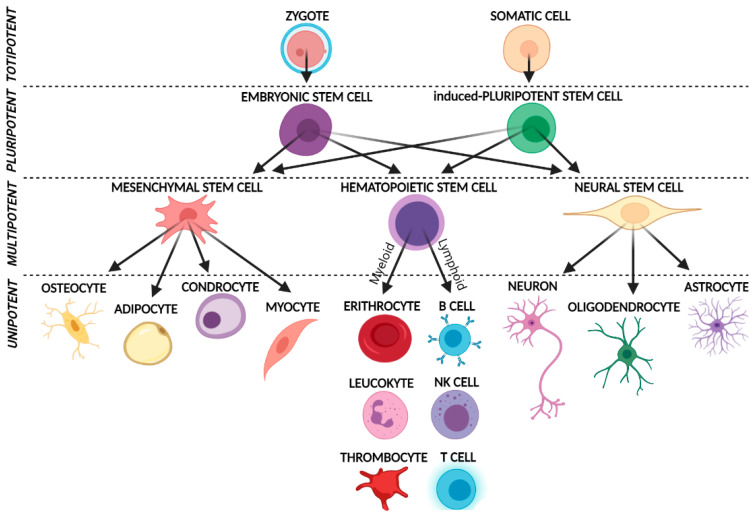
Graphical representation of stem cells based on their specification and differentiation potential.

**Figure 2 biomedicines-11-01587-f002:**
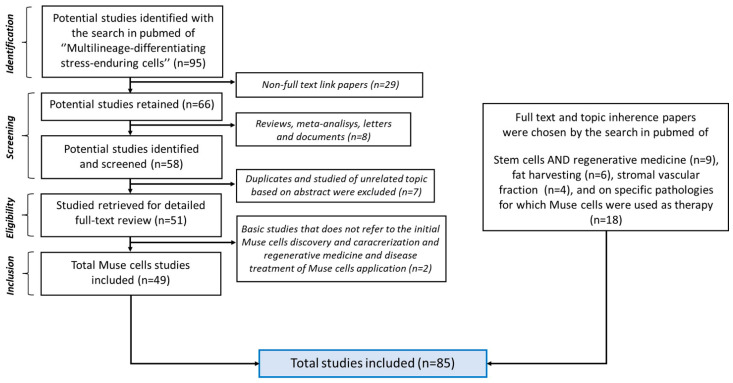
PRISMA flow diagram. After the initial inclusion-exclusion criteria, the total of included papers was 85.

**Figure 3 biomedicines-11-01587-f003:**
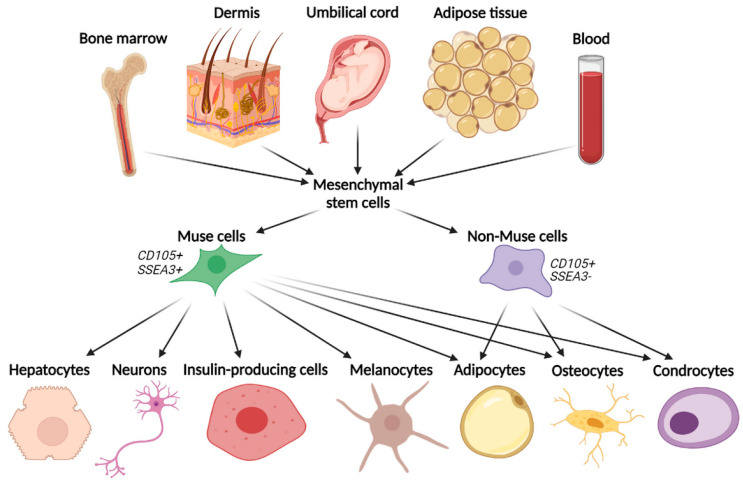
Graphical depiction of tissue origin and differentiation capacity of Multilineage–differentiating stress–enduring cells (Muse cells).

**Figure 4 biomedicines-11-01587-f004:**
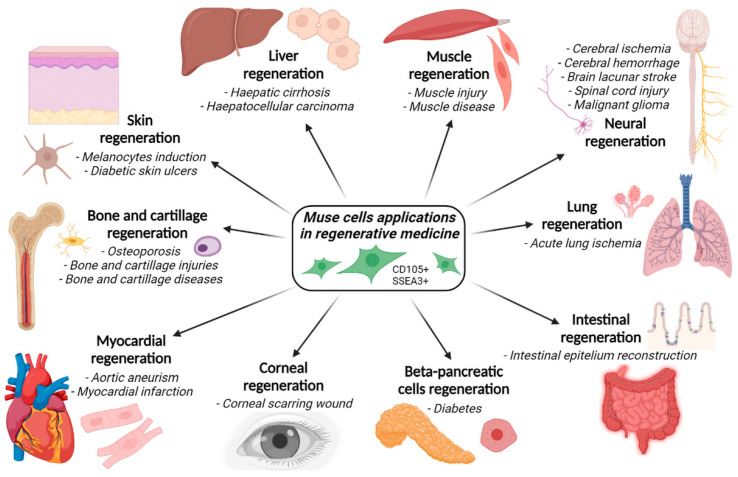
Graphical depiction of Muse cells application in a preclinical model of regenerative medicine.

**Figure 5 biomedicines-11-01587-f005:**
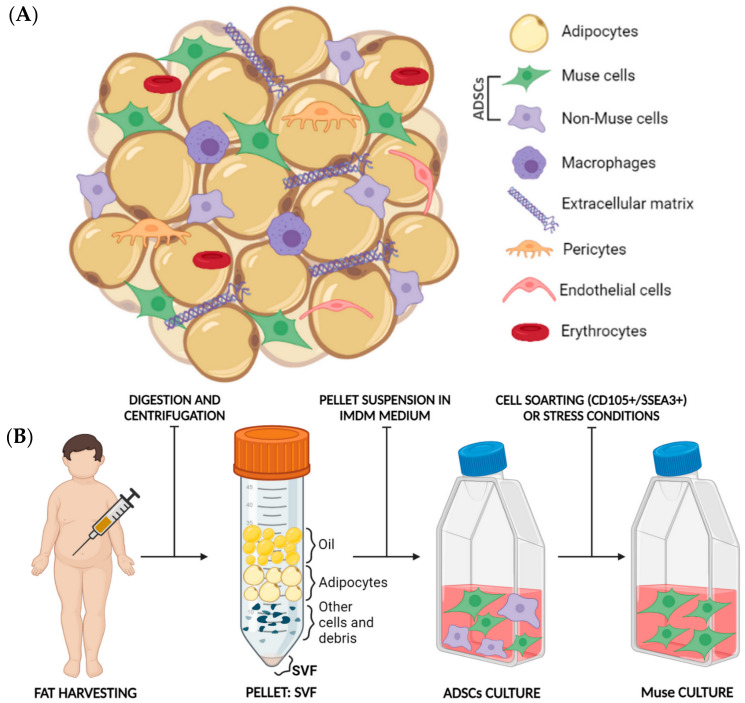
(**A**) Graphical depiction of the different components present in adipose tissue. Fat graft-derived adipose tissue is composed of adipocytes, adipose-derived stem cells (composed of Muse and non-Muse cells), monocytes and macrophages, extracellular matrix components (such as collagen), pericytes, endothelial cells, and erythrocytes. (**B**) Graphical depiction of the Muse cells isolation from lipoaspirates. The fat graft-derived adipose tissue is first digested and centrifugated in a falcon, forming an upper oil layer, followed by an adipocyte layer, a cell debris layer, a protein layer, and from the pellet, which is the stromal vascular fraction (SVF). SVF is then cultured with an ADSCs-specific growth medium (DMEM + 10%FBS), and subsequently, Muse cells are isolated from non-Muse cells via cell sorting (using double-positive CD105+/SSEA3+ cells) or by culture under stress conditions. Muse cells are finally available for further studies and applications.

## Data Availability

All data relevant to this study are included in the manuscript.

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
