# Peer review of "Multilineage-Differentiating Stress-Enduring Cells (Muse Cells): An Easily Accessible, Pluripotent Stem Cell Niche with Unique and Powerful Properties for Multiple Regenerative Medicine Applications"

_biomedicines, 2023, doi:10.3390/biomedicines11061587_

Round 1

Reviewer 1 Report

Overall, this review is generally well-written and nicely illustrated. Multilineage-differentiating stress-enduring (Muse) cells represent a relatively recent take on what's really a much older, and many would say discredited, line of research. This reviewer believes Muse cells remain an interesting and potentially very valuable area of research; however, the authors should include a discussion of regarding the history of such work and acknowledge its controversies and criticisms. For example, Verfaillie's ~2001 work on "multipotent adult progenitor (MAP) cells" generated a great deal of high-profile interest that ultimately didn't progress as expected. There are also explanations such as exosomes carrying various reprogramming factors, which should be discussed. Generally, understanding and acknowledging the history of this field would allow the reader to weigh the evidence regarding this potentially paradigm shifting field. 

Well-written and clear. Minor editing may improve slightly

Author Response

The authors appreciate the reviewer suggestions. It was improved the criticisms and limitations in clinic traslational Muse cells applications in the ''conclusion'' section (the added part is highlighted in yellow), considering additionally topics to be better investigated (such as exosomes or other released factors) and that could partecipate to the regenerative mechanisms of these cell subpopulation. 

Reviewer 2 Report

The review introduced the properties, preparation, and application of Muse cells, which is well-organized and readable. One minor suggestion is that the PRISMA flow diagram should be added.

Author Response

Authors thanks the reviewer for suggestions. It was included a PRISMA flow diagram in the section ''Bibliographic research'' of the paper which helps the reader to better understand the process followed throughout the study. (the added part is highlighted in yellow)